# Physical Activity Interventions in People with Diabetes: A Systematic Review of The Qualitative Evidence

**DOI:** 10.3390/healthcare12141373

**Published:** 2024-07-09

**Authors:** Mireia Vilafranca-Cartagena, Aida Bonet-Augè, Ester Colillas-Malet, Antònia Puiggrós-Binefa, Glòria Tort-Nasarre

**Affiliations:** 1Department of Nursing, Faculty of Health Sciences at Manresa, Universitat de Vic—Universitat Central de Catalunya (UVic-UCC), Av. Universitària 4-6, 08242 Manresa, Spain; ecolillas@umanresa.cat (E.C.-M.); apuiggros@umanresa.cat (A.P.-B.); 2Epi4health Research Group, Faculty of Health Sciences of Manresa, Universitat de Vic—Universitat Central de Catalunya (UVic-UCC), Av. Universitària 4-6, 08242 Manresa, Spain; 3Althaia Foundation, C/Dr Joan Soler 1-3, 08243 Manresa, Spain; 4Department of Nursing and Physiotherapy, University of Lleida, 25002 Lleida, Spain; aida.bonet@udl.cat (A.B.-A.); gloria.tort@udl.cat (G.T.-N.); 5Health Education, Nursing, Sustainability and Innovation Research Group (GREISI), University of Lleida, 25002 Lleida, Spain; 6Transformative Innovation and Simulation Research Group, Faculty of Health Sciences of Manresa, Universitat de Vic—Universitat Central de Catalunya (UVic-UCC), Av. Universitària 4-6, 08242 Manresa, Spain; 7AFIN Research Group and Outreach Centre, Autonomous University of Barcelona, 08193 Bellaterra, Spain

**Keywords:** diabetes mellitus, type 2, exercise, patient compliance, qualitative evidence, review

## Abstract

Objectives: The aim of this systematic review is to summarize the results of qualitative research into people with type 2 diabetes mellitus (T2DM) and their propensity to engage in physical activity (PA), and to identify and analyse their experiences and opinions of interventions and programs designed to increase their adherence to PA recommendations. Design: Systematic review of qualitative studies extracted from databases using the SPIDER systematic search method. The review included studies that combined qualitative and mixed methods research and compiled the experiences and opinions of people with T2DM who had participated in interventions to increase their levels of PA. A thematic summary of qualitative data was performed. Results: The review comprised nine studies published between 2017 and 2021, which included 170 people. Four themes and ten subthemes were identified. The four themes include: (1) factors related to PA, (2) factors related to the program, (3) factors related to the support the participants received and (4) factors related to the person. Conclusion: The support patients receive, both from family/friends and from health providers, is key to consolidating changes in habits and in promoting individualized health education. Future interventions and health policies should reinforce programs designed to promote PA that prioritize the experiences of people in order to increase their adherence to PA programs.

## 1. Introduction

Type 2 diabetes mellitus (T2DM) is a chronic disease that has become a major public health concern both in Spain and worldwide. According to data from the International Diabetes Federation (IDF) and the WHO (World Health Organization), 537 million adults (aged between 20 and 79 years) all over the world live with diabetes: that is, one in every 10 people. This number is expected to rise to 643 million in 2030, and to 783 million in 2045. In Europe, 61 million people (one in 11 adults) live with diabetes, and the number is forecast to reach 67 million in 2030 and 69 million in 2045 [1,2]. In Spain, according to the diabetes.es study [3], the prevalence of type 2 diabetes is 14%, which means that it affects approximately six million people in this country. A million more people in Spain have T2DM but remain undiagnosed, which means 2.3% of the population. T2DM can have serious health consequences and significantly reduces quality of life.

Along with diet and correct adherence to pharmacological treatment, physical activity (PA) is an important tool in the prevention and management of diabetes. Increasing PA and reducing sedentary behavior are an integral part of maintaining glycemic control [4]. The American Diabetes Association recommends that adults with T2DM perform 150 min of moderate-to-vigorous intensity PA per week [5], and breaking up prolonged bouts of sitting with light walking activity has been shown to improve glycemic control [6].

Exercise interventions target resistance (i.e., strength and power), aerobic work, balance, and flexibility [7].

In the primary care setting, PA interventions for people with T2DM include assessment of their current levels of PA, education regarding its benefits, exercise planning, progress monitoring, and referral to a specialist in PA if needed.

Most programs, including face-to-face interventions and those that use technological applications, evaluate effectiveness in the short/medium term (3-12 months) [8,9,10], but not in the long term.

Many adults with T2DM continue to have difficulty complying with recommended guidelines. Numerous interventions are currently underway to increase levels of PA and to bring down levels of sedentary behavior in adults with T2DM, but adherence to the proposed guidelines and the perception of the need to increase PA remains low [11].

In order to understand the factors that affect adherence to PA, it is essential to record people’s experiences and opinions. It is necessary to identify the obstacles that discourage its practice and the facilitators that motivate people to participate, and to define effective strategies for increasing adherence. Furthermore, it is crucial to establish the factors that favor the maintenance of these interventions in the long term. There is little literature documenting the experiences of people with T2DM who have participated in an intervention to increase FA adherence. Many interventions are carried out but adherence remains low. This is why this review will help guide future interventions of this style. The opinions expressed by patients can help health professionals adjust interventions and adapt them to individual needs and preferences and thus ensure adherence to PA programs in the longer term, improving both their general health and their quality of life.

The guiding question of this review is: what are the experiences and opinions of people with T2DM participating in interventions designed to increase adherence to PA?

The aim of this systematic review is to summarize the results of previous qualitative research in this area and to identify and analyse the experiences and opinions of people with T2DM who have participated in interventions intended to increase their level of PA.

## 2. Materials and Methods

After a rigorous selection of qualitative studies on T2DM and PA, a systematic review and thematic synthesis was carried out. The conclusions of the qualitative studies reflect the experiences of the groups that participated in PA interventions and may help to guide the design of more appropriate and effective programs in the future. The knowledge acquired from the synthesis of qualitative studies allows the exploration of statistical heterogeneity in ways that are difficult to study with other methodologies [12]. The present systematic review was conducted in line with the enhanced transparency of reporting qualitative synthesis statement (ENTREQ) [13].

The search strategy included combinations of keywords for each of the four concepts of interest: (1) Type 2 Diabetes Mellitus; (2) exercise or physical activity; (3) perception or experiences or design of qualitative studies; and (4) program or intervention.

The SPIDER scheme [14], described below, was applied in the search for qualitative studies:Sample: adults with T2DMPhenomenon of interest: experiences and opinions of people with T2DM regarding sedentary behavior and PA who have carried out interventions to increase their PADesign: interview, focus group, observation or case studyEvaluation: experiences, opinions, attitudes, beliefs, knowledge and feelingsResearch type: qualitative and mixed methodology.

Qualitative studies were extracted from SCOPUS, Web of Science, CINAHL, PubMed, Cochrane from the inception of each database until December 2022. Studies included in the review were those applied qualitative and mixed research methods to record the experiences and opinions of people with T2DM who have participated in interventions to increase PA.

The electronic search strategy used in PubMed, for example, was: Diabetes Mellitus, Type 2”[Mesh]) AND “Perception”[Mesh]) OR “Life Change Events”[Mesh]) AND “Exercise”[Mesh]) OR “Sedentary Behavior”[Mesh]) OR “Sitting Position”[Mesh]) AND “Clinical Trial” [Publication Type]) AND “Qualitative Research”[Mesh]) NOT “Diabetes Mellitus, Type 1”[Mesh]) NOT “Diabetes, Gestational”[Mesh].

Figure 1 presents the selection of the articles.

The following inclusion and exclusion criteria were applied.

The inclusion criteria were adults with T2DM, qualitative or mixed studies (with a qualitative part), studies that report experiences and opinions of the interventions or programs carried out to increase PA in people with T2DM and studies published in English or Spanish.

The exclusion criteria were people with type 1 or gestational DM.

The search results were imported into Mendeley desktop. Duplicate and unrelated studies were removed. Two raters independently reviewed the study titles and abstracts for eligibility. In all cases, the decision to include or exclude a study was approved by two raters [15], who have experience both in publishing qualitative methodology articles and in publishing reviews (MVC and GTN).

PRISMA guidelines were followed for the performance of this review (Appendix A).

The review process involved two authors who independently reviewed each study based on a checklist in order to reach consensus.

The Spanish Critical Evaluation Skills Program (CASPe) [16] was used to assess the quality of the studies (Table 1). CASPe evaluates qualitative research primarily on the basis of the rigor of the design, the credibility of the results and their relevance to existing practice. To be included, each study must meet the minimum objective criteria, that is, having a score equal to or lower than 4 (total scores on this scale range from 0 (best) to 10 (worst)).

A standardized data collection form (including country, number and type of people, methodology, data analysis, main findings) was produced in advance to compile the data. Subsequently, two authors extracted the data independently.

The use of thematic synthesis [26] identified patterns and key concepts present in the data through an iterative and reflective coding process:

(1)Familiarization with the studies included, through reading the articles several times and taking notes on relevant themes, concepts, and categories.(2)Creation of a coding system by identifying common patterns and themes across the studies included.(3)Search for common themes and patterns among studies, grouping information according to previously defined categories.(4)Data extraction and synthesis, comparing and contrasting the findings from the different studies in order to identify similarities and differences.(5)Interpretation of the findings with respect to the research question, and presentation of general explanations and conclusions.

## 3. Results

### Characteristics of the Studies Included

An initial sample of 665 studies was obtained and after removing the ones that did not meet the inclusion criteria, the final sample comprised 9 studies published between 2017 and 2021 [17,18,19,20,21,22,23,24,25] (Table 2).

A total of 170 individuals were included in the nine studies that finally made up the review. Analytical themes were identified. All nine studies met the CASPe quality evaluation criteria. Eight studies were qualitative and one was mixed [18].

All studies were in English and were carried out in Indonesia, Scotland, the US, South Africa, Belgium, New Zealand, Sweden, Denmark, and Bangladesh. Most of the studies focused on the use of a specific device or application to promote active lifestyles [18, 19, 21, 23-25]. Two studies described a weekly PA program [17,22] and a single study used a visual projection method to understand the meaning of lifestyle change [20].

Four themes and ten subthemes were identified from the experiences and opinions of people with T2DM participating in an intervention to increase adherence to PA (Table 3 shows a summary of the results obtained in the analysis). The PA-related factors analyzed included facilitators and barriers to PA and sedentary behavior. Program-related factors included evaluation of programs and perception of improvement. Support-related factors included family and social support and monitoring. Person-related factors included motivation, perception of improvement, habits, and inner resistance.

### 3.1. PA-Related Factors

#### 3.1.1. Facilitators (Five Studies)

Physiological and psychological benefits motivated people to continue doing PA [17,22,24], as well as the personal desire to improve [19]. Physical exercise was considered a good way to keep diabetes under control: “Walking is the main medicine for diabetes” [25].

#### 3.1.2. Barriers (Seven Studies)

Barriers preventing people from increasing their PA levels or joining PA programs were classified as:

Physiological barriers, such as inability to perform PA, pain, weakness, injuries or diseases associated with T2DM [17,19,22,25]. “In fact, I would like to exercise more; However, sometimes it’s too painful for me, since the doctors told me that I have osteoarthritis and gout in my knees” [22].

Psychological barriers, such as lack of enjoyment when doing PA, lack of motivation, lack of time, lack of knowledge about appropriate types of PA for people with T2DM, lack of confidence in their ability to do the exercises and lack of positive clinical results [17,21].

Social barriers, such as going on vacation, the loss of a loved one, time scheduling, social, occupational or family responsibilities, lack of time and Ramadan [17,22,24]. “I often don’t have enough time to exercise; In fact, I even decided to retire from my job to take care of my elderly mother who needs my support almost constantly” [17]. 

Environmental barriers, such as bad weather, long distances, poor transportation options for reaching the venue of the PA program, unsafe or uneven pedestrian paths, and lack of safe public places [17,19,22,25] “I can’t drive and, unfortunately, my house is quite far away from this hospital” [17].

Barriers related to the program or device, such as the perception that it is complicated, or difficult to set up; text messages that are repetitive and predictable, lack of personalization, lack of interactivity of text messages, technical problems and aversion to text message automation [18,19]. “I can tell that they were a can of limited messages that were repeated over and over again. It seemed like it was coming out and was automated, almost like an alarm clock, that would give you a text message.” [19].

#### 3.1.3. Physical Activity and Sedentary Behavior (Seven Studies)

People reported that they exercised regularly, at least once a week: most walked on their own [17,18,22,24,25], cycled [18], climbed stairs, and did breathing exercises [17]. 

The two studies that talked about sedentary behavior reported that people were unaware of the amount of time they spent sitting during the day [21,23]. 

### 3.2. Factors Related to the Program

#### 3.2.1. Evaluation of the Program (Eight Studies)

Most people were familiar with PA programs and had positive perceptions of them. There was an increase in awareness of the need for PA and people considered the personal approach was very positive [17,19-21].

In the study by Hodgson et al. [18] people discussed the need to integrate all the elements of T2DM healthcare, including PA, into a single support package. Other respondents felt that the program did not provide new information, or that the information offered was scarce [17]. Others stated that the program was easy to use [21] and stressed the importance of personalized follow-up [23,25].

Among the activities, “walking” appealed to almost all the informants in the intervention group because it was familiar to them and was “aligned with what they could see themselves doing” [24]. 

#### 3.2.2. Perceptions of Improvement (Seven Studies)

People approved of the program because it promoted PA [18]; the sense of obligation to follow the prescription of regular walking helped them to modify their behavior [22], and they were motivated by the idea of progress [24]. The program was also easy to adapt to their everyday activities, and this encouraged some people to monitor their changes [21]. The improvements in their physical health and self-reported clinical outcomes also increased their confidence [20]. 

The physical and mental rewards made them feel more energetic and alert and they experienced less mental fatigue [23]; this motivated them to continue doing exercise [17].

### 3.3. Factors Related to the Support the People Received 

#### 3.3.1. Family and Social Support (Six Studies)

Support from family, friends, diabetic support groups, and community walking groups was considered important for PA [18]: “Seeing my friend in exercise class always makes me happy and helps me forget about my problems at home” and people who engaged in independent activities, such as walking, enjoyed the opportunity to socialize with peers [17]: “I feel encouraged to walk since I made friends there [during the walk]. I meet everyone as I walk and we talk about various topics, including our diabetes. I came to know many things about others during walking” [25].

The most frequently mentioned social influence was that of the family. Regular walking provided opportunities to spend quality time with family members, for example, with children or grandchildren. Family support was experienced as something positive, though not crucial [23]. The absence of the family was stressed by some people who lived alone or lacked the normative and social influences provided by the family [22]. Several people mentioned that they joined the program with a family member and experienced valuable social support while engaging in it [21].

#### 3.3.2. Support and Follow-Up (Personalization) (Six Studies)

The environment of trust generated by health professionals encouraged participation in PA [24], creating the feeling that someone cared about them and took an interest in their health problems [25]. The diabetes nurse took on the role of coach, helping the people reflect on their sedentary behavior and on ways to reduce it, to the point that the people did not want to break the agreement they had made with her [23]: “There were a few times when, let’s say I didn’t walk. They would be like: try doing this type of exercise. They would give me an example of what other exercises I could have done or stuff like that. So it opens up your mind to see what other things you can do.” [19].

People stated that the continued support was of great value to them in their attempts to change their behavior, and felt an overwhelming feeling of gratitude. They were well aware of the difficulty of dealing with diabetes and therefore valued the interest shown in them and the time spent supporting them. People felt free to ask questions and to express their fears and concerns [20].

In contrast, the lack of support for PA and the impact on people’s health was highlighted as a concern. Some people felt that the advice on PA given as part of their T2DM care was limited, and that more time was spent discussing medication, blood glucose readings, and diet. As a result, one suggestion made was that their data be downloaded into the system’s database and that a health practitioner should be available to provide personalized feedback and FA analysis: “I think It would be nice if my diabetes nurse could access my data and then we could sit down and talk about how to improve things or just keep me active.” They would like health practitioners to provide additional personalized support for PA focused on their particular exercise and lifestyle needs. Other suggested improvements included more discussion time, further analysis of current activity levels, and the design of detailed exercise programs. “Developing a personalized training package for patients. Further support in the form of specific activity sessions could be established within medical centres … This support could also include information about the benefits of PA for T2DM.” [18].

### 3.4. Factors Related to the Person

#### 3.4.1. Motivation (Seven Studies)

Reinforcing the establishment of a routine was a key motivation. Improved self-esteem and feelings of well-being were associated with the physical act of walking as frequently as other benefits, such as weight loss, improved digestion, improved sleep quality, or improved control of blood glucose [22]. Recording and reviewing daily activity on devices increased levels of activity [19], highlighting the motivational and goal-setting impact: “I was taking more steps than I thought, and this certainly motivated me to get out more.” Achieving the proposed goals produced a sense of accomplishment for users: “I set my own little goals. Setting these goals and reaching them made me feel good about myself.” [18] and “I knew what I did the day before and I was like ·how do I challenge myself to go a little farther, to push myself a little bit hard today?” “But just getting those text messages every day and seeing on the computer how well I was doing, it just gave me more encouragement to do the right stuff” [19]. Confidence regarding diabetes self-management increased over time [20].

Follow-up was crucial: “When for some reason I miss my walking routine, say for 1 or 2 days, many people ask me: ‘where have you been?’, in a way that makes me happy because it means that many people care for me” [17]. People were proud of becoming more active and many reported gaining self-confidence and having fewer feelings of shame or guilt. They felt that it was important to make themselves feel proud and that it would be disappointing if they did not achieve their goals [23]. 

All informants felt able to participate in the PA routines presented in the program and were motivated by the idea of progress. The ability to improve their fitness levels, weight and well-being was the main motivational factor for participating in PA. Achievement facilitates motivation and the desire to continue. Acquiring knowledge and practical experience was motivating for some of the people, particularly for those without prior experience in structured PA [24].

#### 3.4.2. Habits (Seven Studies)

Commitment was seen as an essential factor in sustaining behavior change after the completion of the programs/interventions. Some patients stated that they would not have been able to establish structures for themselves to consolidate the change in their PA habits [24]. Meeting other people and going on walks increased the commitment and was accompanied by a growing need for autonomy and a sense of choice [24,25]. Once the habit is acquired, it becomes second nature and taking exercise becomes instinctive [19,22], even more so if individuals are committed to participating in lifestyle changes and to deciding what kinds of activities to choose and how to carry them out [23].

The interventions/programs contributed to positive lifestyle changes, which were triggered by regular conversations about diabetes-related topics [20]. Patients felt that they had to keep their promises since they would be evaluated at the following session, and this helped them adopt healthy lifestyle habits [21]. Small but continuous changes made it easier to reduce the amount of sitting time [23].

#### 3.4.3. Inner Resistance (Three Studies)

Some people felt they had to overcome their own internal barriers. Lack of progress and unmet goals were devastating for motivation toward continuing [24]. People wanted to avoid feelings of shame and guilt. They were unaware of the time they spent sitting, and they reported changing their behavior step by step after overcoming their initial inner resistance to increasing their daily steps or using sit-stand desks [23].

For some people, adhering to the PA prescription was considered a low priority. In these cases, competing interests or demands were deemed more important than following the walking prescription [22]. One people said that he himself was the factor for not doing PA: “I think it is indeed a good program, but the factor is only me. I just don’t have an urge to join it." [17].

## 4. Discussion

### 4.1. Factors Related to PA (Facilitators, Barriers and Sedentary Behavior)

Performing PA increases confidence and psychological well-being [27]. Additionally, the desire to be healthy and to keep the disease under control is also a facilitator for engaging in PA [28].

People with T2DM present particularly difficult challenges for physicians, such as their marked heterogeneity, the potential presence of multiple comorbidities, the physiological changes of the aging process, greater susceptibility to hypoglycemia, greater dependence on care, and frailty [29,30,31]. Performing PA at night is a barrier identified in several studies and can be overcome by recommending PA that can be carried out within the home, for example by using personal gym equipment [32]. Scheduling PA such as walking may help to reduce this barrier to initiating and performing PA, is accessible, and does not require much time [33]. Several authors have reported lack of motivation as a major barrier [34].

In the study by Rozas [35], 87% of patients were sedentary and did not know how long they spent sitting during the day. In contrast, in the study by Hashim [36], people were aware of their sedentary behavior.

### 4.2. Factors Related to the Program (Evaluation of the Program and Perception of Improvement)

Many programs and interventions are currently underway to improve the habits of people with T2DM, although adherence to healthy lifestyles remains low. Nonetheless, recent studies have reported that participation in online diabetes health communities encourages self-management [37] and that activities designed to promote PA are effective, at least in the short term [38]. Telephone follow-up and the individualization of programs can increase the effectiveness of interventions, even if they are group-based [39]. Balducci et al. reported that a behavioral intervention strategy resulted in a sustained increase in PA and a decrease in sedentary time compared to standard care [40]. 

Programs or devices designed to increase PA must be easy to use, especially since almost half of people with T2DM are aged 65 years or older [29]. Indeed, in our study we found that easy-to-access and easy-to-use programs raised awareness and behavior change. Archundia-Herrera et al. stressed the importance of explicit, individualized, applicable, realistic and practical information [41]. In line with our results., in the study by Nielsen [42], people reported concerns about continuing to exercise on their own after the end of the intervention, testimonies that underline the need for professional and social support.

### 4.3. Factors Related to the Support the People Received (Family and Social Support, and Monitoring)

Social influence (that is, the influence of both friends and family) can help to increase the disposition to perform PA [43]. Personalized PA behavior change interventions increase activity [44]. The study by Yin [45], which evaluated a peer support program focusing on metabolic and behavioral parameters in people with T2DM, confirmed the importance of continuous support. In that study, peer supporters used a checklist to review adherence to medication, diet, exercise, and glucose control; the authors found that, after four years, continuous peer support had improved people’s self-care behavior, including performance of PA, psychological health, and glycemic control.

Nursing staff have an important role to play in the education of people with with T2DM and should be adept at offering personalized information [46,47] as we have identified in our study, but according to a study by Jabardo-Camprubí [48], the adoption of PA is a priority for most nurses but few dedicate time to encouraging adherence.

### 4.4. Factors Related to the Person (Motivation, Habit, Inner Resistance)

Motivation has been shown to be a facilitator for carrying out PA. According to self-determination theory, people perform exercise because it provides pleasure and satisfaction (intrinsic motivation); they may integrate exercise in their value system or they may simply see it as being important for their health. Interactions between health providers and patients and the lifestyle interventions proposed should focus on promoting self-motivated reasons for change and on integrating change into the personality [49]. Identification and acceptance of a new lifestyle leads to greater adherence [50]. Motivators to exercise are strengthened by positive experiences of exercise [51] and by the desire to increase wellbeing [52], in line with our results. Interactions that raise motivation should be promoted so as to favor self-care with respect to diet and exercise [38].

## 5. Conclusions

This systematic review has identified and analysed the experiences and opinions of people with T2DM participating in interventions designed to increase their adherence to PA.

After analyzing factors related to PA, program characteristics, support received, and individual factors, we conclude that the support that patients receive both from family/friends and from health systems, and the provision of individualized health education to increase motivation and promote commitment (whether individually or via group interventions/programs) are key to consolidating the lifestyle changes proposed.

The practical recommendations to take into account for the development of PA programs that seek the adherence of patients with T2DM can be develop comprehensive support systems involving family, friends, and healthcare providers to encourage and sustain PA adherence, the individualized health education, the accessible and flexible PA programs, and the behavioral and psychological support that provide regular feedback and monitor progress to keep patients motivated and aware of their improvements.

Future measures and health policies should reinforce interventions aimed at increasing PA and should pay particular attention to people’s experiences in order to increase their adherence to PA programs. Our findings highlight the importance of individualizing these interventions to cater to the unique needs and circumstances of each patient.

This article has limitations. First, as only studies published in English or Spanish were included, certain articles that met the other inclusion criteria may have been overlooked. Second, the terms “exercise” and “physical activity” are used interchangeably in the literature, creating some confusion when comparing studies. As regards the studies included, not all of them considered sedentary behavior in the design of interventions to improve the lifestyle of people with T2DM; similarly, although they described many interventions/programs for improving PA adherence in people with T2DM, few offer qualitative analyses.

## Figures and Tables

**Figure 1 healthcare-12-01373-f001:**
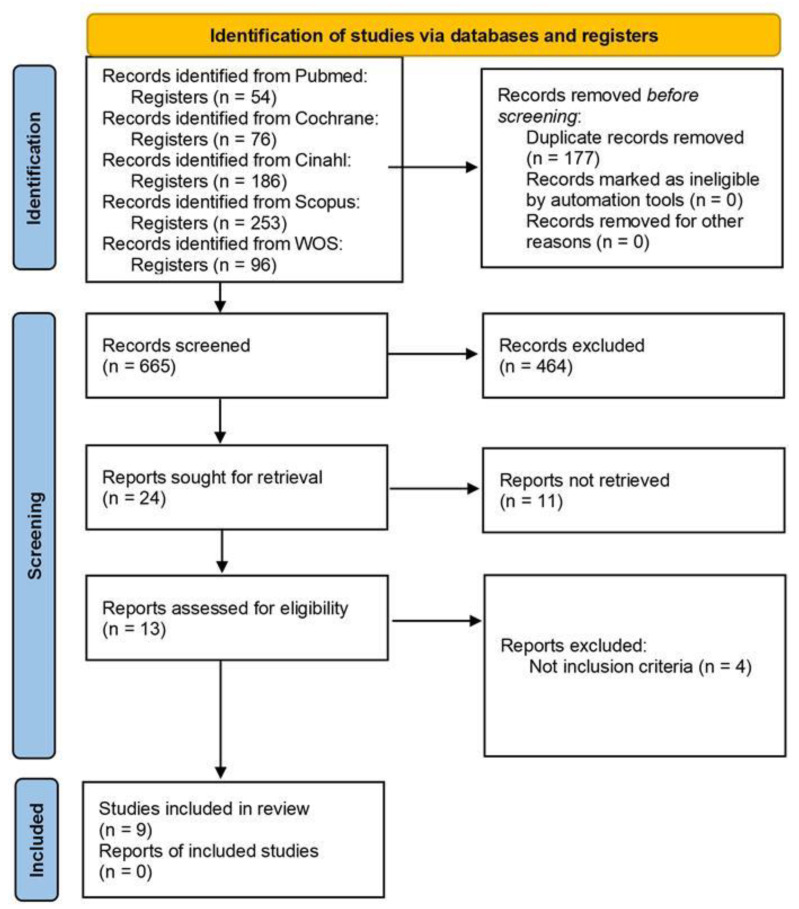
Flowchart representing the selection of studies.

**Table 1 healthcare-12-01373-t001:** CASPe (Risk of bias, validity and methodological quality. Total scores on this scale range from 0 (best) to 10 (worst)).

Details of Study	Type of Study	Were the Objectives of the Study Clearly Defined?	Is the Qualitative Methodology Adequate?	Is the Research Method Appropriate to Achieve the Objectives?	Was the Strategy for Selecting People Consistent with the Research Question and the Method Used?	Were the Data Collection Techniques Consistent with the Research Question and the Method Used?	Was there a Reflection on the Relationship between the Researcher and the Research Object (Reflexivity)?	Were Ethical Aspects Taken into Account?	Was the Data Analysis Rigorous Enough?	Is The Presentation of the Results Clear?	Are the Research Results Applicable?	Number of Points Obtained
Arovah N et al. 2019 [17]	Qualitative	0	0	0	0	0	1	0	1	0	0 Indonesia	1
Hodgson W et al. 2021 [18]	Mixed (with a qualitative part)	0	0	0	0	0	2	0	1	0	0 Scotland	3
Horner G et al. 2017 [19]	Qualitative	0	0	0	0	0	0	0	0	0	0 US	0
Pienaar M et al. 2021 [20]	Qualitative	0	0	0	0	0	2	0	1	0	1 South Africa	4
Poppe et al. 2018 [21]	Qualitative	0	0	0	0	0	1	0	1	0	0 Belgium	2
Reynolds et al. 2020 [22]	Qualitative	0	0	0	0	0	0	0	0	0	0 New Zealand	0
Syrjälä et al. 2021 [23]	Qualitative	0	0	0	0	0	1	0	1	0	0 Sweden	2
Walker K et al. 2018 [24]	Qualitative	0	0	0	0	0	1	0	1	0	0 Denmark	2
Yasmin F et al. 2020 [25]	Qualitative	0	0	0	0	0	1	0	1	0	1 Bangladesh	3

Legend: Score 0 = Yes; Score 1 = Don’t know; Score 2 = No.

**Table 2 healthcare-12-01373-t002:** Characteristics of the studies included.

Author(s), Year of Publication	Population	N	Methodology (Phenomeno-Logical, Descriptive, etc.)	Data Collection (Structured or Semi-Structured Interviews)	Type of Analysis	Intervention	Principal Findings Lack of Knowledge Environmental Restrictions…
Arovah N et al. 2019 [17]	Adults with T2DM from the public hospital of Yogyakarta, Indonesia	28 people, 62.8 ± 5.4 y	?	4 focus groups	Thematic analysis	Weekly physical activity program for T2DM of 60 min or more	Physical activity pattern Perceived facilitators of physical activity Perceived barriers to physical activity Perceptions of physical activity programs
Hodgson W et al. 2021 [18]	Adults with T2DM, Scotland	12	Mixed	Semi-structured interviews	Qualitative, abductive thematic analysis	Study of the use of Fitbit activity trackers (device to support an active lifestyle in people with T2DM) for the 4-week quantitative element of the study	Current provision of physical activity advice within care for T2DM Integrated elements of type 2 diabetes health care Security and data management Barriers to Fitbit use Personalization of physical activity support for type 2 diabetes Using Fitbit as a motivation tool Goal setting and Fitbit features preferred by active users
Horner G et al. 2017 [19]	US	31 (51.4 y)	Grounded theory approach	Two focus groups and telephone interviews at 6–12 months	Thematic analysis	The Text to Move (TTM) program was a randomized controlled trial using text messages for education, reminders, and motivational messages	Effect of participation in the study Effect of using pedometer: motivation through display of step count and related texts Effect of text messages; text messages as informative or useful for generating ideas to increase physical activity Barriers to the effectiveness of the intervention; text messages were too repetitive and predictable. Lack of personalization, Lack of interactivity of text messages.
Pienaar M et al. 2021 [20]	South Africa	12 (women 51–84 y)	Narrative inquiry	Mmogo-method (similar to focus groups); people construct their responses with indigenous material	--	Support intervention that used the Mmogo-method^®^, a visual-based narrative inquiry. TNB Diabetes Peer Support Intervention	Positive lifestyle changes. They adjusted their diets and improved their activity levels. Ongoing support. They expressed an overwhelming sense of gratitude. They acknowledged that community health workers provided them with important support to continue with behavior change. Greater confidence and sense of connection. People reported that their confidence regarding diabetes self-management increased over time. Confidence was due to improvements in physical health and self-reported clinical outcomes.
Poppe et al. 2018 [21]	Ghent Hospital, Belgium	21 adults with T2DM (57–81 y)	?	Semi-structured interviews	Targeted content analysis	MyPlan 2.0’ is a self-regulation-based eHealth intervention (website) targeting physical activity and sedentary behavior. ‘MyPlan 2.0’ users can choose between the “increase in physical activity” and “decrease in sedentary lifestyle” modules.	Usefulness of the website. It does not provide new information; they were already well informed, but it did raise awareness that they needed to change their behavior. They indicated that they were not aware of the amount of time they spent sitting during the day and considered it interesting to obtain information. Those who were already active did not find the website relevant. The website encouraged some people to monitor their changes and helped them evaluate their plan, but others believed it was superfluous. Website design. Personal approach. Short, understandable questions, and the website was easy to navigate. People liked the short duration of each session Knowledge. While most people were well aware of the beneficial effects of an active lifestyle, some questioned whether it also applied to them, as they did not feel any change in themselves by being more active. Social support. Several people mentioned that they visited the website with a family member and experienced social support while doing so.
Reynolds et al. 2020 [22]	Dunedin, New Zealand	28 adults with T2DM (age 60 ± 9 y)	Grounded theory	Interviews	Thematic analysis	Prescription walking for at least 10 min after breakfast, lunch and dinner, every day for 3 months	Motivators: importance of routine and benefits (once you get into the habit, walking becomes instinctive); the support of their families and regulation of behavior. Participation in the study (“the study has motivated me to walk and do some exercise more than I normally would”) Barriers: walking at night (included feelings of insecurity in the dark or a preference for sedentary behavior), time (not walking in rain or extreme temperatures), discomfort (pain from walking), and behavioral regulation competing priorities (I just want to go watch TV, it couldn’t fit into my life)
Syrjälä et al. 2021 [23]	Sweden	15 adults with T2DM	?	Semi-structured interviews	Content analysis with inductive approach	3-month intervention that included mHealth, activity tracker (Garmin Vivofit3) and SMS reminders, an initial person-centered face-to-face counseling session, and three follow-up phone calls from a diabetes nurse specialist in PA.	1. From baby steps to milestones, reflecting three categories: 1.1 “Small changes make it easier to reduce the amount of sitting time” 1.2. ”Encouraged by a trusted coach” 1.3.”Physical and mental rewards matter” 2. Adaptation strategies that fit me and my workplace, reflecting four categories: 2.1. “It depends on me” 2.2.”Take advantage of support” 2.3. “Use creativity to find practical solutions to interrupt sitting” 2.4. “Meet expectations”
Walker K et al. 2018 [24]	Copenhagen, Denmark	5 (41–70 y). Interviews before, during and after the intervention	Self-determination theory (induction-deduction): investigate motivation factors for physical activity	Semi-structured interviews on 3 occasions (1 year)	--	Rehab program InterWalk (InterWalk app-dictated interval walking training pace based on an individual’s walk test)	1. Balance the need for commitment and a sense of choice: 1.1. Commitment fosters physical activity 1.2. Physical activity challenged by other commitments 1.3. Transfer of commitment to a new structure 2. Feel competent and experience progress: 2.1. Commitment encourages physical activity 2.2. Physical activity challenged by other commitments 2.3. Transfer of commitment to a new structure 2.4. Feeling able to make a difference 2.5. Achievement facilitates motivation 2.6. Knowledge about what works and proof that it matters 3. Minor theme: setting, environment, and actual activity (theme covered the meaning of setting and activity on motivation for physical activity): 3.1. Perceived suitability for participation 3.2. Perceived suitability for participation 3.3. Current activity
Yasmin F et al. 2020 [25]	Dhaka, Bangladesh	18 (30–79 y)	?	Semi-structured interviews (30–40 min)	Deduction/induction	m-Health mobile health project (call center and interactive voice calls)	1. Perception of m-Health 2. Life with diabetes 3. Management of hospital visits and services 4. Management of medication intake 5. Practical diet 6. Physical exercise. More than half were already performing PE, family and social environment beneficial; physical problems such as weakness, palpitations, and pain as the main reasons for irregular physical exercise, and also lack of time and unfavorable weather conditions. Unsafe and uneven pedestrian paths or a lack of safe public places (e.g., a park) for walking were also mentioned as a barrier to regular walking. The people stated that they walked at home (in the hallway or on the terrace) due to their physical problems and/or unfavorable outdoor conditions. Ramadan was also cited as a factor for irregular physical exercise due to the potential risk of an attack of hypoglycemia and general weakness resulting from prolonged fasting. But the majority of people reported that they continued with physical exercise (walking), mainly in the morning, to avoid hypoglycemia at night while fasting. 7. Political situation of the country during the study.

**Table 3 healthcare-12-01373-t003:** Summary of the results.

Themes	Subthemes	Results
PA-related factors	Facilitators	Physiological and psychological benefits
	Barriers	Physiological barriers, pain, weakness, injuries or diseases, lack of time, lack of knowledge, social barriers, environmental barriers, and barriers related to the program or device
	Physical activity and sedentary behavior	People were unaware of the amount of time they spent sitting during the day.
Factors related to the program	Evaluation of the program	Positive perceptions, but it needs improvements.
	Perceptions of improvement	The physical and mental rewards made them feel more energetic and alert and they experienced less mental fatigue.
Factors related to the support the people received	Family and social support	Support from family (most frequently mentioned), friends, diabetic support groups, and community walking groups was considered important for PA.
	Support and follow-up (personalization	Environment of trust was generated by health professionals and continued support, but in contrast, the lack of support for PA and the impact on people’s health was highlighted as a concern.
Factors related to the person	Motivation	Reinforcing the establishment of a routine follow-up, motivation by the idea of progress
	Habits	The interventions/programs contributed to positive lifestyle changes.
	Inner resistance	Internal barriers, lack of progress, and unmet goals. For some people, adhering to the PA prescription was considered a low priority.

## Data Availability

Not applicable.

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
