# Peer review of "Physical Activity Interventions in People with Diabetes: A Systematic Review of The Qualitative Evidence"

_healthcare, 2024, doi:10.3390/healthcare12141373_

Round 1

Reviewer 1 Report

Comments and Suggestions for Authors

First of all, I would like to thank the authors for the contribution they are making since, as all professionals who are dedicated to the prescription of physical activity know, the biggest problem we encounter is adherence to it and studies that focus on it. in searching for the reasons why this phenomenon takes place is of high scientific value.

Likewise, I would like to give some recommendations to the authors, because although I believe they have done a very beneficial task for science, certain aspects must be improved for publication.

In general, it seems like a careful text to me, but it is true that perhaps a better organization of the contents could prevent concepts and information from being repeated and make the reading lighter. I recommend the authors read the article again and find a less cumbersome way to explain their methods and results.

On the other hand, take advantage of the reading to solve errors in the references or typographical errors, such as the way of referencing in this line: p. 14, L234.

Now I will give some recommendations section by section.

INTRODUCTION

The introduction seems to me to have been written as if it were a summary of a theoretical framework, that is, instead of providing a reasoned justification of the need to develop the study due to the lack of evidence in this regard or the lack of similar evidence or the need for updating etc. I see that it has focused more on telling concepts that all of us interested in the subject know. It will therefore be important to focus the introduction more, in my opinion, on the reasons why a qualitative review of the published evidence in this regard is important.

On the other hand, a more relevant contribution would be on p.1 L.36 when talking about T2DM it does not say that the M refers to the concept mellitus. I think that needs to be corrected.

METHODS

I think that the wording of the main objective of the study should be at the end of the introduction, not at the beginning of the methods, as well as in general the first two paragraphs.

I recommend that the authors follow the PRISMA recommendations as they say they follow in order to better structure the sections and ensure that they follow a logical order and do not repeat information.

I think that the inclusion and exclusion criteria will be better explained in a paragraph than in bullet points.

RESULTS

The results in general seem very interesting to me but it seems to me more like a reading of the tables that are presented than really a summary of the evidence that was intended. Perhaps creating a summary or a table with the main results or a figure, that is, something more visual, could facilitate their interpretation.

DISCUSSION

It seems to me to be the biggest handicap of the review presented. I find the discussion to be a brief summary of the results, which obviously it should be, but on the other hand I miss a more critical view of the results, a reasoning from the authors based on evidence that argues why what happens happens and that does not focus only on telling what happens.

On the other hand, I would put the limitations within this section, and not as a separate section.

And finally, given the exhaustive analysis that is carried out, I miss something like "practical recommendations to take into account for the development of PA programs that seek the adherence of patients with T2DM."

Thank you so much,

Author Response

Dear Reviewer 1:

Comment: First of all, I would like to thank the authors for the contribution they are making since, as all professionals who are dedicated to the prescription of physical activity know, the biggest problem we encounter is adherence to it and studies that focus on it. in searching for the reasons why this phenomenon takes place is of high scientific value. Likewise, I would like to give some recommendations to the authors, because although I believe they have done a very beneficial task for science, certain aspects must be improved for publication. In general, it seems like a careful text to me, but it is true that perhaps a better organization of the contents could prevent concepts and information from being repeated and make the reading lighter. I recommend the authors read the article again and find a less cumbersome way to explain their methods and results.

On the other hand, take advantage of the reading to solve errors in the references or typographical errors, such as the way of referencing in this line: p. 14, L234.

Author’s reply: We have fixed this bibliographic error.

Comment: Now I will give some recommendations section by section.

INTRODUCTION

Comment: The introduction seems to me to have been written as if it were a summary of a theoretical framework, that is, instead of providing a reasoned justification of the need to develop the study due to the lack of evidence in this regard or the lack of similar evidence or the need for updating etc. I see that it has focused more on telling concepts that all of us interested in the subject know. It will therefore be important to focus the introduction more, in my opinion, on the reasons why a qualitative review of the published evidence in this regard is important.

Author’s reply: We have justified the reasons why this review is important.

Comment: On the other hand, a more relevant contribution would be on p.1 L.36 when talking about T2DM it does not say that the M refers to the concept mellitus. I think that needs to be corrected.

Author’s reply: We have modified it.

METHODS

Comment: I think that the wording of the main objective of the study should be at the end of the introduction, not at the beginning of the methods, as well as in general the first two paragraphs.

I recommend that the authors follow the PRISMA recommendations as they say they follow in order to better structure the sections and ensure that they follow a logical order and do not repeat information.

Author’s reply: We have modified it.

Comment: I think that the inclusion and exclusion criteria will be better explained in a paragraph than in bullet points.

Author’s reply: We have explained the inclusion and exclusion criteria in a paragraph.

RESULTS

Comment: The results in general seem very interesting to me but it seems to me more like a reading of the tables that are presented than really a summary of the evidence that was intended. Perhaps creating a summary or a table with the main results or a figure, that is, something more visual, could facilitate their interpretation.

Author’s reply: We have already added a table with the summary results.

DISCUSSION

Comment: It seems to me to be the biggest handicap of the review presented. I find the discussion to be a brief summary of the results, which obviously it should be, but on the other hand I miss a more critical view of the results, a reasoning from the authors based on evidence that argues why what happens happens and that does not focus only on telling what happens.

Author’s reply: We had already added quite a few critical articles but the wording was not appropriate or was not clear enough, we have modified it.

Comment: On the other hand, I would put the limitations within this section, and not as a separate section.

Author’s reply: We have included limitations on the conclusions.

Comment: And finally, given the exhaustive analysis that is carried out, I miss something like "practical recommendations to take into account for the development of PA programs that seek the adherence of patients with T2DM."

Author’s reply: We have added this part to the conclusions.

For more details, please see the revised version of the manuscript.

Thank you so much,

Reviewer 2 Report

Comments and Suggestions for Authors

Peer Review Report for the manuscript Physical Activity Interventions in People with Diabetes: 2 A Systematic Review of the Qualitative Evidence. The manuscript described a systematic review of qualitative information on factors focused on participants’ perspectives on improving physical activity among individuals with type 2 diabetes. The review found four themes and 11 subthemes. From those, support from family and healthcare providers is considered the most influential factor in improving physical activity behaviors among individuals with T2D.

The topic interests Healthcare readers amid the steadily increasing prevalence of T2D worldwide. The manuscript is well-written and organized. I have several comments for minor revision and hope the authors will consider them.

1. One overarching suggestion is to rephrase “patients” to “individuals” or “people,” e.g., individuals or people with type 2 diabetes, and use consistently throughout the paper, either individuals or people.

2. Another suggestion is to use “participants” consistently for the participants. Twice in the paper, the word “informant” was used for participants, lines 239 & 316.

3. Lines 26 – 28, suggest re-constructing the sentence for a clearer meaning. For example, Four themes and 11 subthemes were identified. The four themes include: (1) …, (2) …, (3) …, and (4) … Also, I suggest revising “factors related to support” on line 28 to “factors related to the support the participants received, and …

4. Line 38, the abbreviation WHO must spell all the words out. 

5. Line 41, … “the figure” is forecast … I suggest replacing the word “figure” with “number” (perhaps).

6. Line 42, di@bet.es must be cited appropriately.

7. Line 44, “It” has also been… What is “It”? I suggest rephrasing. Also, I suggest the authors provide the rate of undiagnosed diabetes.

8. Line 107 – 109, I suggest rephrasing for a more apparent meaning. For example, “Studies included in the review were those applied qualitative…”

9. Lines 106 – 116: I do not understand how the authors extracted articles from the search engines and also manually searched for them on the same search engine. I hope the authors will concisely combine the two paragraphs for better understanding.

10. Line 137, I suggest the authors provide the two reviewers’ credentials and expertise to ensure a qualified preliminary review and exclude.

11. Line 142, I suggest listing the “minimum objective criteria.”

12. For the results section, I suggest the authors revise the order of theme and subtheme for consistency. For example:

i. 3.1 Characteristics of the Studies

ii. 3.2 Themes and Subthemes à 3.2.1. Physical Activity related factors; 3.2.1.1. Facilitator; … ;

13. For the Discussion Section, I suggest the authors arrange the themes in the same order as the Result Section.

14. I suggest the authors discuss the findings integrating the four themes to rationale the conclusion of … lines 409 – 412.

Congratulations to the authors for the job well done.

Author Response

Dear Reviewer 2:

Comment: Peer Review Report for the manuscript Physical Activity Interventions in People with Diabetes: 2 A Systematic Review of the Qualitative Evidence. The manuscript described a systematic review of qualitative information on factors focused on participants’ perspectives on improving physical activity among individuals with type 2 diabetes. The review found four themes and 11 subthemes. From those, support from family and healthcare providers is considered the most influential factor in improving physical activity behaviors among individuals with T2D.

The topic interests Healthcare readers amid the steadily increasing prevalence of T2D worldwide. The manuscript is well-written and organized. I have several comments for minor revision and hope the authors will consider them.

  1. Comment: One overarching suggestion is to rephrase “patients” to “individuals” or “people,” e.g., individuals or people with type 2 diabetes, and use consistently throughout the paper, either individuals or people.

Author’s reply: We have changed that.

  1. Comment: Another suggestion is to use “participants” consistently for the participants. Twice in the paper, the word “informant” was used for participants, lines 239 & 316.

Author’s reply: We have changed that.

  1. Comment: Lines 26 – 28, suggest re-constructing the sentence for a clearer meaning. For example, Four themes and 11 subthemes were identified. The four themes include: (1) …, (2) …, (3) …, and (4) … Also, I suggest revising “factors related to support” on line 28 to “factors related to the support the participants received, and …

Author’s reply: We have changed that.

  1. Comment: Line 38, the abbreviation WHO must spell all the words out. 

Author’s reply: We have added all the words.

  1. Comment: Line 41, … “the figure” is forecast … I suggest replacing the word “figure” with “number” (perhaps).

Author’s reply: We have replaced that.

  1. Comment: Line 42, di@bet.es must be cited appropriately.

Author’s reply: We have cited that appropriately.

  1. Comment: Line 44, “It” has also been… What is “It”? I suggest rephrasing. Also, I suggest the authors provide the rate of undiagnosed diabetes.

Author’s reply: We have rephrasing that.

  1. Comment: Line 107 – 109, I suggest rephrasing for a more apparent meaning. For example, “Studies included in the review were those applied qualitative…”

Author’s reply: We have changed that.

  1. Comment: Lines 106 – 116: I do not understand how the authors extracted articles from the search engines and also manually searched for them on the same search engine. I hope the authors will concisely combine the two paragraphs for better understanding.

Author’s reply: We've unified it and explained it better.

  1. Comment: Line 137, I suggest the authors provide the two reviewers’ credentials and expertise to ensure a qualified preliminary review and exclude.

Author’s reply: We have specified the credentials.

  1. Comment: Line 142, I suggest listing the “minimum objective criteria.”

Author’s reply: We have listed the minimum objective criteria.

  1. Comment: For the results section, I suggest the authors revise the order of theme and subtheme for consistency. For example:
  2. 3.1 Characteristics of the Studies
  3. 3.2 Themes and Subthemes à 3.2.1. Physical Activity related factors; 3.2.1.1. Facilitator; … ;

Author’s reply: We have changed that.

  1. Comment: For the Discussion Section, I suggest the authors arrange the themes in the same order as the Result Section.

Author’s reply: We have arranged the themes in the same order as the Result Section.

  1. Comment: I suggest the authors discuss the findings integrating the four themes to rationale the conclusion of … lines 409 – 412.

Author’s reply: We integrated the themes.

Comment: Congratulations to the authors for the job well done.

Author’s reply: Thank you very much for your valuable comments that have increased the rigor of this work. We hope that the changes made are as expected.

For more details, please see the revised version of the manuscript.